# Updates on the CDK4/6 Inhibitory Strategy and Combinations in Breast Cancer

**DOI:** 10.3390/cells8040321

**Published:** 2019-04-06

**Authors:** Navid Sobhani, Alberto D’Angelo, Matteo Pittacolo, Giandomenico Roviello, Anna Miccoli, Silvia Paola Corona, Ottavia Bernocchi, Daniele Generali, Tobias Otto

**Affiliations:** 1Department of Medical, Surgery and Health Sciences, University of Trieste, Piazza Ospitale 1, 34129 Trieste, Italy; farmacia.micoli@gmail.com (A.M.); sil.corona@hotmail.it (S.P.C.); ottavia.b88@gmail.com (O.B.); dgenerali@units.it (D.G.); 2Department of Biology and Biochemistry, University of Bath, Bath BA2 7AY, UK; 3Department of Orthopedics and Orthopedic Oncology, University of Padova, 35128 Padova, Italy; matteo.pittacolo@studenti.unipd.it; 4Department of Internal Medicine III, University Hospital RWTH Aachen, 52074 Aachen, Germany

**Keywords:** cyclin-dependent kinases, cyclin-dependent kinase 4 and 6 inhibitors, targeted therapies, breast cancer

## Abstract

Breast Cancer (BC) is the second most common type of cancer worldwide and displays the highest cancer-related mortality among women worldwide. Targeted therapies have revolutionized the way BC has been treated in recent decades, improving the life expectancies of millions of women. Among the different molecular pathways that have been of interest for the development of targeted therapies are the Cyclin-Dependent Kinases (CDK). CDK inhibitors are a class of molecules that already exist in nature and those belonging to the Cyclin dependent kinase inhibitors family INK4 that specifically inhibit CDK4/6 proteins. CDK4/6 inhibitors specifically block the transition from the G1 to the S phase of the cell cycle by dephosphorylation of the retinoblastoma tumor suppressor protein. In the past four years, the CDK4/6 inhibitors, palbociclib, ribociclib, and abemaciclib, received their first FDA approval for the treatment of Hormone Receptor (HR)-positive and Human Epidermal growth factor Receptor 2 (HER2)-negative breast cancer after showing significant improvements in progression-free survival in the PALOMA-1, MONALEESA-2 and the MONARCH-2 randomized clinical trials, respectively. After the encouraging results from these clinical trials, CDK4/6 inhibitors have also been investigated in other BC subtypes. In HER2-positive BC, a combination of CDK4/6 inhibitors with HER2-targeted therapies showed promise in preclinical studies and their clinical evaluation is ongoing. Moreover, in triple-negative BC, the efficacy of CDK4/6 inhibitors has been investigated in combination with other targeted therapies or immunotherapies. This review summarizes the molecular background and clinical efficacy of CDK4/6 inhibitors as single agents or in combination with other targeted therapies for the treatment of BC. Future directions for ongoing clinical trials and predictive biomarkers will be further debated.

## 1. Introduction

With the advent of personalized therapies, more molecular targets have been discovered—the expression of which leads to tumor proliferation in solid tumors such as breast cancer. The discovery of such molecular mechanisms became of fundamental importance for pharmaceutical companies aiming to create more accurate therapies with the aim of targeting cancer cells selectively. Among such molecular targets, the cyclin/cyclin-dependent kinase (CDK) holoenzyme was discovered nearly 30 years ago.

Later, this molecule was observed to be a benchmark for several cell cycle transitions and, over the next 10 years, the first non-specific CDK inhibitor was used in clinical practice. Partially because of dose-limiting toxicity that hinders their use, these pan-CDK inhibitors showed poor results [1]. Currently, more specific compounds are available, which selectively inhibit CDK4 and CDK6 while exhibiting lower toxicity. CDK4/6 targeting is now under evaluation in a large number of clinical trials, both as single agents as well as in combination with inhibitors of other signalling pathways. Therefore, it is of paramount importance to obtain a better understanding of the mechanisms behind their anti-tumor activity.

CDK4/6 activity links the cell cycle to several different extracellular signalling pathways [2] (Figure 1). Most normal cells (considered non-transformed and non-immortal cells) restrict themselves irreversibly into the G1/G0 phase of the cell cycle and this process relies on the activity or the phosphorylation status of the Retinoblastoma tumor suppressor protein (Rb). This protein has well-known growth suppressive properties, which are mostly linked to its ability to bind the transcription factors of the *E2F* gene family, thereby curbing the transcription of target promoters [3,4,5]. Once Rb is in a hyper-phosphorylated status, its interaction with E2F and other transcription factors is impaired. In normal cells, the Rb phosphorylation status is generally a result of the consecutive action of CDK4 or CDK6 kinases together with D-type cyclin subunits, followed by cyclin E–CDK2 complexes [6,7]. In addition, the expression of cyclins and CDK inhibitors such as p27^Kip1^, p21^Cip1^, and p16^Ink4a^ are regulated by extracellular signals, causing positive or negative regulation of CDK activity, respectively. More specifically, Cyclin-Dependent Kinase Inhibitor 1B (p27^Kip1^) and Cyclin-Dependent Kinase Inhibitor 1A (p21^Cip1^) have been reported to inhibit CDK2 kinases, and Cyclin-Dependent Kinase Inhibitor 2A (p16^Ink4a^) to inhibit CDK4/6 kinases [2] (Figure 1).

In most human cancer cells, this pathway is impaired due to either the loss of Rb itself, overexpression of cyclin D1, mutation of CDK4 (causing insensitivity to CDK inhibitors) or the loss of p16^Ink4a^ [4]. This can not only affect the response of cancer cells to extracellular signals but also circumvent the requirement for the sequential order of phosphorylation by CDKs during the inactivation of Rb. For this reason, CDK2 may not be essential in some cancer cells.

Rb, together with other proteins that have a pivotal role in controlling the commitment decision, has already been reported for its non-cell cycle-related activity, even though Rb is the main cell cycle target of CDK4/6 [8,9]. On the other hand, CDK4/6 kinases can also promote the phosphorylation of factors involved in cell differentiation, metabolism and apoptosis [10,11]. Where applicable, possible non-Rb targets have been included since these might be involved in the senescence-promoting, immunological or metabolic outcomes correlated with these drugs.

The aim of this review is to summarize the molecular background and the latest evidence on the efficacy of CDK4/6 inhibitors for the treatment of breast cancer. The potential use of these drugs in combination with other targeted therapies and the role of predictive biomarkers will be debated.

## 2. The Rb Pathway

In response to various external stimuli, a number of signaling pathways (e.g., PI3K–AKT–mTOR, MAPK, STAT, NF-kB, Wnt–β-catenin, ER, PR, and AR) induce the expression and stability of D-type cyclins such as cyclin D1. D-type cyclins then form a complex with CDK4 or CDK6, thereby promoting progression of the cell cycle from G1 phase to S phase by inactivating tumor suppressors Rb, Retinoblastoma-like protein 1 (p107), and Retinoblastoma-like protein 2 (p130) (Figure 1). This complex is stabilized by proteins such as p21^Cip1^ [2]. Once activated by D-type cyclins, the CDK4 and CDK6 mono-phosphorylate Rb. This phosphorylation of the Rb protein partially relieves the Rb-mediated suppression of the family of E2F transcription factors, enabling the expression of cyclin E, which is another S-phase cyclin. Cyclin E subsequently binds to and activates CDK2, which then hyper-phosphorylates Rb, further liberating E2F transcription factors and enabling the expression of a wide range of genes promoting the transition from G1 to S phase [12]. This cell cycle transition triggers a commitment to complete the cell cycle and thereby allows cancer cell growth.

## 3. Mechanism of Action of CDK4/6 Inhibitors

Cyclin-dependent kinase inhibitors include naturally occurring proteins, such as those belonging to the INK4 family—namely p16^Ink4a^ [13], p15^Ink4b^ [14], p18^Ink4c^ [15,16], and p19^Ink4d^ [16,17] (Figure 1). This family of proteins specifically inhibits the catalytic activity of CDK4 and CDK6 [2,18]. INK4 proteins consist of multiple ankyrin repeats and bind only to CDK4 and CDK6. In contrast, the CDK interacting protein/kinase inhibitory protein family (CIP/KIP)—p21^Cip1^ [19,20,21,22,23], p27^Kip1^ [24,25,26], and p57^Kip2^ [27,28]—contain very similar motifs within their amino-terminal moieties endowing them with the capacity to bind to both cyclin and CDK subunits, leading to a variety of effects depending on the type of cyclin–CDK complex [29,30,31,32,33]. For instance, while they inhibit the kinase activity of CDK2, CDK4 and CDK6 complexes are not inhibited but rather stabilized. Inhibitors of CDK4/6 block the cell cycle at the G1-to-S transition phase [18]. They do so by “switching-off” the kinase activity, triggering the dephosphorylation of Rb, resulting in a block of cell cycle progression in mid-G1 phase. Consequently, they lead to an arrest of the cell cycle and prevent the further proliferation of cancer cells. Importantly, CDK4/6 activity is commonly elevated in cancer cells. For instance, overexpression of cyclin D1 occurs frequently in breast cancer [34,35,36,37,38]. This, in turn, increases the activity of CDK4/6 and hence, makes the G1-to-S transition checkpoint an interesting therapeutic target. Consequently, selective and reversible inhibitors of CDK4/6 activity such as palbociclib, ribociblib and abemaciclib block the cell cycle at the G1 phase and thereby prevent cancer progression. Loss of function of p16^Ink4a^ (encoded by *CDKN2A*) or Rb (encoded by *RB1*) is generally mutually exclusive in cancer cells and each one promotes tumorigenesis. Mutation or hypermethylation of *CDKN2A* cause a loss of inhibition of CDK4/6 by p16^Ink4a^, eventually leading to an increase in pro-mitotic signalling. Similarly, when the *RB1* gene is mutated, cyclin E1 and CDK2 become constitutively activated, initiating the transcription of pro-mitotic genes. In the latter scenario, it has been initially postulated that cancer cells become independent from the CDK4/6 pathway and that CDK4/6 inhibition would be ineffective. Therefore, both p16^Ink4a^ and CDK4/6 are capable of exerting their cell cycle function only if Rb is completely functional [39] (Figure 1). Hence, selection of patients on the basis of their Rb mutational status gained interest based on the possibility of predicting the efficacy of CDK4/6 inhibitors in cell lines. However, the same conclusion was not reproducible in clinical settings so far and more biomarkers are needed, as will be discussed in the biomarkers section.

## 4. FDA-Approved CDK4/6 Inhibitors in Advanced or Metastatic ER-Positive Breast Cancer

CDK4/6 inhibitors proved to be beneficial in both preclinical and clinical trials for ER-positive breast cancer, especially when combined with anti-estrogen therapies [40,41]. Palbociclib and ribociclib received their first FDA approval in February 2015 and March 2017 after positive results from the PALOMA-1 [42] and MONALEESA-2 [43] studies, respectively, for the first-line treatment of ER-positive, HER2-negative advanced breast cancer for postmenopausal women in combination with anti-estrogen therapy. Abemaciclib received its first FDA approval in September 2017, following the MONARCH-1 trial [44], as a single agent for women and men with ER-positive, HER2-negative advanced BC after disease progression following endocrine therapy and prior chemotherapy. At the same time, following the positive results of the MONARCH-2 trial [45], abemaciclib also received approval in combination with fulvestrant for women with ER-positive, HER2-negative advanced BC after disease progression following endocrine therapy. These three CDK4/6 inhibitors have different toxicities, dosing schedules, pharmacokinetics, and potencies. Of note, to each one of these drugs, there are toxicity profiles that must be taken into consideration. These have been demonstrated to change when given as monotherapy or in combinations with other drugs, as summarized in Table 1. The clinical development of these three drugs for ER-positive HER2-negative BC will be discussed in the following paragraphs.

Palbociclib (IBRANCE; PD0332991; Pfizer; C_24_H_29_N_7_O_2_) is an orally available drug with a low enzymatic IC_50_ of 11 nM for CDK4 and 15 nM for CDK6. It was the first of a series of CDK4/6 inhibitors, whose anti-cancer potency was proved for breast cancer cells. In fact, Finn and colleagues showed that there is a synergy between this drug and tamoxifen in ER-positive breast cancer cell lines [41]. Moreover, they uncovered that palbociclib was able to enhance the sensitivity of ER-resistant cell lines to tamoxifen by inhibiting Rb phosphorylation in these cells [41]. Additionally, the ability to inhibit cancer growth even for tamoxifen-resistant tumors was demonstrated in vivo using patient-derived xenograft mouse models (PDX) [52]. The first phase I study using palbociclib in patients with solid tumors showed a safety profile favourable for proceeding to the next phases of clinical development, with myelosuppression being the main dose-limiting toxicity (DLT) [53]. In contrast to what is usually observed for chemotherapy, the neutropenia observed by the inhibition of CDK4/6 is associated with a very low risk of febrile neutropenia [42]. In the current clinical practice, palbociclib is administered at a starting dose of 125 mg daily for three consecutive weeks, followed by one week off, in combination with endocrine therapy. This drug was initially investigated as monotherapy in a phase II, single-arm study for metastatic breast cancer patients expressing Rb protein in their tissues, with the primary objective to evaluate tumor response and with some of the secondary objectives including progression-free survival (PFS) and expression of Rb, p16^Ink4a^ loss, and cyclin D1 gene (*CCND1)* amplification [46]. However, since these patients were heavily pretreated, modest results were detected regarding the objective response rate (ORR), clinical benefit rate (CBR), and median PFS (mPFS), which were 5%, 19%, and 3.7 months, respectively [46]. Additionally, no biomarkers for predictive sensitivity to the drug were detected [46]. On the other hand, a subsequent randomized phase II clinical trial called PALOMA-1/TRIO-18 showed that the combination of letrozole with palbociclib showed a remarkable improvement in the median PFS of 20.2 months compared to 10.2 months with letrozole alone (HR, 0.49, *p* = 0.0004) in ER-positive HER2-negative BC patients receiving this treatment as first-line therapy [42]. After these encouraging results, a phase III randomized clinical trial including 666 postmenopausal women, termed PALOMA-2, was conducted to confirm these promising results. It showed that the median PFS improved using the combination of letrozole with palbociclib to 24.8 months compared to 14.5 months in the control arm with letrozole and placebo (HR, 0.58; 95% CI, 0.46 to 0.72; *p* < 0.001) [47]. In addition, palbociclib has also been investigated for BC patients who had progressed on previous endocrine therapy. For instance, in a randomized phase III clinical trial called PALOMA-3, including 521 patients with HR-positive HER2-negative breast cancer randomly assigned to receive either palbociclib with fulvestrant or placebo with fulvestrant, the median overall survival (OS) improved by 6.9 months in the palbociclib arm versus the control arm, at a median follow-up duration of 44.8 months (HR, 0.81; 95% CI, 0.64 to 1.03, *p* = 0.09) [54]. Although these improvements in OS were not statistically significant, they support the notion that the addition of palbociclib to fulvestrant improves mPFS by 4.9 months, previously reported in 2016 [48].

Moreover, the data proved that there was an even higher improvement in those patients who previously showed a sensitivity to the endocrine therapy and in those patients without visceral disease [55]. In fact, Turner et al. showed that there is an improvement of 6.4 months in patients who were sensitive to endocrine therapy and had visceral disease. The mPFS was 19.3 months with palbociclib plus letrozole vs. 12.9 months in placebo plus letrozole (HR, 0.63; 95% CI, 0.47 to 0.85), while the ORR was 55.1% vs. 40.0%, respectively. Furthermore, they also showed that there is an mPFS improvement of 9.3 months in patients without visceral disease. The mPFS was 16.6 months in patients who received palbociclib plus fulvestrant vs. 7.3 months in patients who received placebo plus fulvestrant [55]. PALOMA-3 is an ongoing phase III, randomized-to-control, double-blind clinical trial. It showed that the addition of palbociclib to endocrine therapy with fulvestrant proved a clinical benefit in mPFS to 9.2 months (95% CI, 7.5 to not-estimable) compared to the 38 months (95% CI, 3.5 to 5.5) in placebo with fulvestrant arm (NCT01942135) [48]. 

Ribociclib (KISQALI; LEE011; Novartis; C_23_H_30_N_8_O) is another orally available drug with a low enzymatic IC_50_ of 10 nM for CDK4 and 40 nM for CDK6. This drug was approved to be used with a starting dose of 600 mg daily for three consecutive weeks and one week off [56]. Regarding toxicity, its profile is very similar to that of palbociclib, with neutropenia and leukopenia being the most frequent adverse events at grade 3/4 as reported in large clinical trials (Table 1) [43]. The phase III randomized clinical study including 668 postmenopausal women with previously untreated ER-positive HER2-negative advanced breast cancer, named MONALEESA-2, showed that the addition of ribociclib to letrozole significantly improved PFS, which at 18 months was of 63% in the ribociclib group and 42.2% in the placebo group (*p* < 0.001) [43]. The results of this trial led to the FDA approval for the use of this drug in combination with aromatase inhibitors as first-line treatment for ER-positive HER2-negative advanced or metastatic BC in March 2017. Most recently, the results from a phase III randomized clinical trial including 672 premenopausal women, named MONALEESA-7, showed that the combination of ribociclib with endocrine therapy (tamoxifen or aromatase inhibitors) significantly improved the median PFS from 13.0 months in the placebo group (endocrine therapy alone) to 23.8 months in the ribociclib group (HR, 0.55; 95% CI, 0.44 to 0.69; *p* < 0.0001) [57]. This led to an expanded FDA approval in July 2018 to include pre/perimenopausal women. Furthermore, ribociclib has also been investigated in combination with fulvestrant in patients that were either untreated or relapsed after endocrine therapy. The phase III randomized clinical study including 726 postmenopausal women with ER-positive HER2-negative advanced breast cancer, named MONALEESA-3, showed that the addition of ribociclib to fulvestrant significantly improved median PFS from 12.8 months in the placebo group to 20.5 months in the ribociclib group (HR, 0.59; 95% CI, 0.48 to 0.73; *p* < 0.001) [50]. This was followed by an FDA approval for this treatment combination in July 2018. Monaleesa-7 is the first phase III clinical trial evaluating a CDK4/6 inhibitor as first-line therapy in premenopausal and perimenopausal women HR+/HER2- MBC. It showed that the ribociclib with either tamoxifen or an NSAI plus goserelin arm lead to a significantly improved PFS of 23.8 months (19.2 months to not-estimable) versus 13.0 months (11.0 to 16.4 months) in the placebo with NSAI/tamoxifen plus goserelin arm (HR, 0.553, *p* = 0.0000000983) [57].

Abemaciclib (VERZENIO; LY2834219; Lilly; C_27_H_32_F_2_N_8_) is the most potent orally available drug with the lowest enzymatic IC_50_, i.e., 2 nM for CDK4 and 10 nM for CDK6. Abemaciclib’s structure enables crossing the blood–brain barrier at lower doses and may remain on-target for a longer time than palbociclib, as proven by orthotopic (intracranial) xenografts of glioblastoma cells [58]. Therefore, it has anti-tumor effects on metastatic lesions affecting the central nervous system (CNS) [59]. Regarding dosing, it has been approved to be given at 200 mg as monotherapy or 150 mg when given with endocrine therapy. The most common adverse event is diarrhoea, which starts during the first week of therapy. Moreover, most patients treated with this drug show an asymptomatic increase of serum creatinine as abemaciclib is a competitive inhibitor of efflux transporters of creatinine in kidneys (e.g., MATE and OCT2) [60]. Randomized phase III clinical trials proved the beneficial effect of abemaciclib in combination with hormonal therapy. In ER-positive HER2-negative advanced BC, both ORR and mPFS significantly improved when used as first-line therapy in 493 postmenopausal women vs. placebo control (HR, 0.54; 95% CI, 0.41 to 0.72; *p* = 0.00002; MONARCH-3) [51]. This combination was hence approved by the FDA in February 2018. Furthermore, in patients who had relapsed after endocrine therapy, a combination of abemaciclib with fulvestrant showed significant benefit compared to fulvestrant alone (median PFS of 16.4 vs. 9.3 months, respectively; HR, 0.55; 95% CI, 0.45 to 0.68; *p* < 0.001; MONARCH-2) [45]. This led to the first FDA approval for abemaciclib in September 2017. In contrast to palbociclib and ribociclib, abemaciclib also showed significant clinical activity as monotherapy in women with ER-positive HER2-negative metastatic BC, who had relapsed after previous endocrine therapy and chemotherapy. In the single-arm phase II study MONARCH-1 including 132 patients, single-agent treatment with abemaciclib exhibited promising clinical activity after 12 months (ORR, 19.7%; 95% CI, 13.3 to 27.5; median PFS, 6.0 months) [44]. Based on these results, the FDA granted approval for this poor-prognosis patient group in September 2017.

After the clinical trial results for CDK4/6 inhibitors that led to the FDA approval of the three compounds, more ongoing clinical studies have been testing CDK4/6 inhibitors either on their own or in combination with other targeted therapies in advanced BC. For instance, an active randomized phase III clinical investigation, called PEARL, is testing palbociclib with endocrine therapy versus chemotherapy (capecitabine) in postmenopausal patients with ER-positive HER2-negative metastatic breast cancer, who showed resistance to aromatase inhibitors (NCT02028507, Table 2). More ongoing and recruiting clinical trials are evaluating the efficacy of CDK4/6 inhibitors in other breast cancer subtypes and after the disease has progressed (Table 2 and Table 3).

## 5. CDK4/6 Inhibitors in HER2-Positive Breast Cancer

Amplification or overexpression of the human epidermal growth factor receptor 2 (HER2/Erbb2) occurs in approximately 15–20% of human breast cancers [61]. In these HER2-positive cancers, the constitutive activation of downstream signalling pathways by HER2 promotes tumorigenesis and metastasis. In recent decades, several HER2-targeted therapies have been approved for the treatment of HER2-positive breast cancer: the monoclonal antibodies, trastuzumab and pertuzumab, the kinase inhibitor, lapatinib and neratinib, and the antibody-drug conjugate, trastuzumab emtansine (T-DM1). While these targeted therapies have improved outcomes, metastatic HER2-positive breast cancer remains incurable as tumors eventually develop therapy resistance. Therefore, understanding the mechanisms of resistance and investigating other treatment options for HER2-positive breast cancer patients remains an important research goal.

While most of the clinical research in the past decade focused on the use of CDK4/6 inhibitors in ER-positive HER2-negative breast cancer, several earlier studies already suggested CDK4/6 inhibition as a potential treatment for HER2-positive tumors. In one of the first studies on the activity of the CDK4/6 inhibitor, palbociclib, Finn and colleagues demonstrated that luminal-type breast cancer cells expressing estrogen receptor, including luminal-type cells with HER2 amplification, were significantly more sensitive to palbociclib than basal-type ER-negative cells [41].

Moreover, palbociclib showed enhanced activity in combination with trastuzumab or T-DM1 in HER2-amplified breast cancer cells in vitro [41,62]. These results were in line with genetic studies in mice that pointed to an essential function of CDK4/6 in HER2-positive breast cancer. For instance, mice lacking cyclin D1 or CDK4, as well as mice expressing mutant cyclin D1 (incapable of activating CDK4/6) were resistant to breast cancer induced by the Erbb2/HER2 oncogene [34,63,64]. Furthermore, the acute deletion of cyclin D1 or inhibition of CDK4/6 kinase activity using palbociclib blocked the progression of HER2-driven breast cancer in mice [18]. Similar to palbociclib, abemaciclib was recently shown to exhibit substantial activity against luminal ER-positive HER2-positive breast cancer cells both in vitro and in xenografts [65]. Its activity was further enhanced in combination with HER2-targeted therapy [65].

Although these studies suggested the use of CDK4/6 inhibitors in HER2-positive breast cancer, they did not investigate mechanisms of resistance to HER2-targeted approaches. In the past, alterations in several proteins and signalling pathways were shown to mediate resistance to HER2-targeted therapies, including the PI3K–AKT pathway and the HER2 receptor itself. More recently, the study by Goel and colleagues investigated mechanisms of resistance using a doxycycline-regulated HER2-overexpressing mouse model for breast cancer [66]. They discovered that tumors recurring after HER2 withdrawal or after trastuzumab treatment overexpressed cyclin D1 and the CDK4 protein in the nucleus, most likely as a result of mutations associated with the hyperactivation of the MAPK pathway. Similar results were obtained by another group who investigated lapatinib-resistant breast cancer cells [62]. Hence, Goel and colleagues hypothesized that cyclin D1 overexpression may mediate resistance to HER2-targeted therapies. Indeed, the overexpression of cyclin D1 was capable of reducing the sensitivity of breast cancer cells to HER2-targeted agents [66]. Of note, tumors resistant to HER2 targeting were dependent on cyclin D1–CDK4 complexes and their growth could be inhibited using the CDK4/6 inhibitor abemaciclib. Importantly, they also showed that the combined targeting of HER2 and CDK4/6 in lapatinib/trastuzumab-resistant HER2-positive breast cancer cells caused not only the additive but synergistic inhibition of cell growth and cell viability in vitro. Strikingly, the combined targeting of CDK4/6 and HER2 using abemaciclib and trastuzumab in patient-derived xenograft (PDX) models of treatment-refractory HER2-positive breast cancer displayed enhanced anti-tumor activity in vivo.

While numerous preclinical studies already suggested the cyclin D–CDK4/6 axis as an attractive target in HER2-positive breast cancer (including cancers resistant to HER2-targeting drugs), clinical studies have mainly focused on ER-positive, HER2-negative breast cancer. Nevertheless, some of these studies also included patients with HER2-positive disease. For instance, a phase II study using palbociclib in Rb-positive advanced breast cancer included two patients with ER-positive HER2-positive breast cancer [46]. One of these patients had a partial response (PR), while the other patient had stable disease (SD) lasting five months. Another study in Japan included one patient with HER2-positive advanced breast cancer, who showed a PR after treatment with abemaciclib [67]. To date, the largest study evaluating CDK4/6 inhibition as novel targeted therapy for patients with HER2-positive advanced breast cancer was published three years ago [68]. In that trial, eleven patients with HER2-positive advanced breast cancer were treated with abemaciclib. Of these, four patients had a PR and seven patients had SD (lasting at least 24 weeks in two patients). This corresponds to a response rate of 36% and a clinical benefit rate (CR + PR + SD ≥ 24 weeks) of 55%. These patients also showed a median progression-free survival of 7.2 months. While this study was not suited to compare the efficacy of CDK4/6 inhibition to HER2-targeted approaches, it indicated a substantial clinical activity of CDK4/6 inhibitors in this patient subgroup.

Currently, a number of clinical phase II and phase III studies are ongoing to evaluate the clinical benefit of combining HER2-targeted approaches with CDK4/6 inhibitors for treating HER2-positive advanced breast cancer. Examples are the PATRICIA study (phase II; palbociclib + trastuzumab ± letrozole; NCT02448420), the PATINA study (phase III; HER2-targeted therapy ± palbociclib; NCT02947685), the monarcHER study (phase II, abemaciclib + trastuzumab ± fulvestrant; NCT02675231) and another phase II study (ribociclib + trastuzumab or T-DM1 ± fulvestrant; NCT 02657343). The results from these studies are expected between 2019 and 2020.

In addition, CDK4/6 inhibitors are also considered in the early stage setting of HER2-positive breast cancer. While the PALTAN study (phase II; palbociclib + trastuzumab + letrozole; NCT02907918) is still ongoing, results from the NA-PHER2 study (NCT02530424) were recently published [69]. In this phase II study, previously untreated HER2-positive ER-positive breast cancer patients received a combination of palbociclib, trastuzumab, pertuzumab, and fulvestrant (an ER antagonist). Out of 30 patients that were assessed, 29 patients (97%) achieved a clinical objective response. Moreover, during the following surgery, a pathological complete response could be confirmed for eight of these patients (27%) [69]. These results suggest a potential use of CDK4/6 inhibitors and HER2-targeted approaches also in the neo-adjuvant setting, possibly replacing neo-adjuvant chemotherapy.

While questions regarding efficacy are still awaiting results from ongoing and future randomized trials, these preclinical and clinical results already demonstrate promise for the use of CDK4/6 inhibitors in this subtype of breast cancer. Future directions may also include defining subsets of HER2-positive breast cancer patients that are most likely to respond to CDK4/6 inhibition. While the presence of functional Rb protein is a well-established marker correlating with response to CDK4/6 inhibition [39], one report suggested that an 11-probe gene expression signature may predict sensitivity to palbociclib [70]. Future studies will be necessary to validate the use of such predictors.

## 6. CDK4/6 Inhibitors in Triple-Negative Breast Cancer

Triple-negative breast cancer (TNBC) is defined by the lack of expression of estrogen and progesterone receptor proteins, as well as the absence of HER2 overexpression or amplification. It accounts for approximately 10–15% of breast cancer cases and comprises a relatively heterogeneous group of tumors [61], most of which are associated with a basal-like gene expression signature [71]. Importantly, TNBC displays poor prognosis compared to other breast cancer subtypes [61]. In addition, about 20% of TNBCs lack functional Rb protein, rendering them insensitive to inhibitors of the cyclin D–CDK4/6 pathway. Therefore, TNBC has not been considered as a promising tumor subgroup for targeted therapies that impinge on the cyclin D–CDK4/6 pathway. However, a number of preclinical results have challenged this dogma in the last two years.

In the study of Asghar and colleagues, the luminal androgen receptor (LAR) subtype of TNBC was identified to be sensitive to CDK4/6 inhibition using palbociclib both in vitro and in a LAR-TNBC xenograft model in vivo [72]. They observed that this sensitivity was associated with a requirement for CDK4/6 activity to re-enter the cell cycle from a quiescent state after completing mitosis. This sensitivity could also be explained by the association of AR expression with Rb expression. Another study reported that sensitivity of TNBC cells towards the CDK4/6 inhibitor abemaciclib was associated with high levels of Rb and phospho-Rb, as well as low levels of p16^Ink4a^ [65]. Of note, abemaciclib was potent enough to induce tumor regression in almost half of the xenografts, which were established using a biomarker-selected TNBC cell line [65]. Interestingly, even though a large fraction of TNBCs may not be sensitive towards CDK4/6 inhibition, palbociclib may still be beneficial by inhibiting the metastasis of TNBC cells to the liver and lung [73].

In addition to the potential use of CDK4/6 inhibitors as monotherapy in sensitive subsets of TNBC, several reports indicated promising preclinical activity in combination with other targeted therapies. A combination of CDK4/6 inhibition (by ribociclib) with PI3Kα inhibition (by BYL719) showed the enhanced inhibition of tumor growth, as well as enhanced tumor immunogenicity and T cell activation in xenograft models of TNBC [74]. Moreover, the combination of CDK4/6 inhibition, PI3Kα inhibition, and immune checkpoint inhibition (anti-PD1 and anti-CTLA4) was able to achieve long-term survival in a syngeneic TNBC mouse model [74]. Other promising combinations with CDK4/6 inhibitors include inhibition of mTOR by MLN0128 [75], as well as targeting of EGFR using erlotinib in a subset of TNBC characterized by high expression of MT4-MMP, EGFR, and Rb [76].

Among the clinical studies that investigated the efficacy of CDK4/6 inhibitors in breast cancer, a few also included patients with triple-negative disease. For instance, a phase II study using palbociclib in Rb-positive advanced breast cancer included four patients with TNBC, and all of them displayed disease progression [46]. Furthermore, a phase I study evaluating the antitumor activity of abemaciclib included nine patients with triple-negative advanced breast cancer [68]. Among these patients, three had SD (lasting at least 24 weeks in one patient), and the median progression-free survival was only 1.1 months. This corresponds to a clinical benefit rate (CR + PR + SD ≥ 24 weeks) of 11%. While these clinical results are disappointing, a few studies are still ongoing. For example, a clinical phase II study is investigating the use of the CDK4/6 inhibitor abemaciclib in patients with Rb-positive TNBC (NCT03130439). Moreover, two clinical phase I/II studies concentrate on the androgen receptor (AR)-positive subtype of TNBC. They investigate the combination of anti-androgen therapy (using bicalutamide) with CDK4/6 inhibition by using either palbociclib (NCT02605486) or ribociclib (NCT03090165).

Another study conducted by Liu et al. evaluated the effect of palbociclib in combination with an anti-androgen enzalutamide in TNBC cells [77]. The combined treatment with enzalutamide in TNBC cells positive for AR/RB may potentiate the cytostatic effect induced by palbociclib. In addition, palbociclib-mediated G1 arrest in AR-positive/RB-proficient TNBC cells was attenuated by RB knockdown. This study indicated that palbociclib in combination with enzalutamide may be a therapeutic strategy for AR-positive/RB-proficient TNBCs [77].

Clearly, future clinical studies that attempt to evaluate the efficacy of CDK4/6 inhibitors in TNBC would greatly benefit from biomarker-guided patient selection to achieve meaningful clinical response rates and to benefit this group of patients. For example, as already mentioned for HER2-positive disease, one report suggested that an 11-probe gene expression signature may predict sensitivity to palbociclib [70]. Furthermore, another study identified a set of Rb/E2F-regulated genes as biomarkers for response to abemaciclib [65]. In addition, based on preclinical data presented above, several combinations of CDK4/6 inhibition with other targeted therapies or immunotherapy might be useful to improve the clinical activity of CDK4/6 inhibitors in TNBC.

## 7. CDK4/6 Inhibitor Combinations with Other Targeted and Immune Therapies

### 7.1. CDK4/6 and PI3K-mTOR Inhibitions

There is crosstalk between the CDK4/6 and the PI3K–mTOR pathways, resulting in a good rationale for testing the combined inhibition of the two pathways in order to block tumor growth [66,78,79,80]. In fact, Goel et al. evinced that inhibition of CDK4/6 not only suppressed Rb phosphorylation, but also reduced the TSC2 phosphorylation, thereby partially hindering mTORC1 activity. In line with this, in vivo results proved that CDK4/6 sensitized PDX to HER2-targeted therapies [71]. Additionally, Vora et al. demonstrated through a screening of 42 CDK4/6 inhibitors that they were able to sensitize *PI3KCA* mutation-bearing breast cancer cell lines to PI3K inhibitors. The strongest sensitizing CDK4/6 inhibitor was ribociclib. Furthermore, they confirmed this data using *PIK3CA* mutation-bearing mouse xenografts, in which co-treatment with CDK4/6 and PI3K inhibitors elicited tumor regression [79]. Moreover, it has been demonstrated that resistance to CDK4/6 inhibitors in ER-positive breast cancer cell lines could be dependent on the activation of a compensatory PI3K non-canonical cyclin D1–CDK2 pathway leading to the phosphorylation of Rb. Using ER-positive BC cell lines, two studies showed in vitro that a combination of treatment with PI3K and CDK4/6 inhibitors can overcome resistance to single-agent CDK4/6 inhibitor, because of the downregulation of cyclin D1 [79,81]. Vora and colleagues uncovered that CDK4/6 inhibitors were able to sensitize *PIK3CA*-mutated cell lines (T47D, MCF7, and MDA-MB-453) to PI3K inhibitors (BYL719 and GDC-0941). The efficacy was then tested in vivo using MCF7R-derived xenograft mouse models, but the results showed only a modest improvement [79]. Subsequently, Herrera-Abreu and colleagues investigated the combination of CDK4/6 and PI3K inhibitors in vitro using MCF7 and T47D, as well as in vivo using patient-derived xenograft (PDX) mouse models [81]. They showed that combining CDK4/6 and PI3K inhibitors triggered cancer cell apoptosis in vitro and in PDX mice. Furthermore, the authors demonstrated that a combination of endocrine therapy, CDK4/6 inhibition, and PI3K inhibition is even more effective in vitro as well as in PDX models, in which disease stabilization and tumor regression were observed [81].

Also in TNBC, a synergistic effect has been observed using both PI3K and CDK4/6 inhibitors. In fact, pre-clinical studies indicated that the combination of palbociclib with taselisib or ribociclib with alpelisib is more efficient compared to each one of the drugs on its own [72,74]. The inhibition of the PI3K signalling pathway increased sensibility of the cancer cells to CDK4/6 inhibitors palbociclib through a mechanism that could be partially attributable to the suppression of post-mitotic CDK2 activity [72]. Based on such encouraging preclinical data, there are current clinical trials testing CDK4/6 and PI3K inhibitors combinations. For ER-positive HER2-negative BC, the combination of ribociclib, fulvestrant, and a PI3K inhibitor (BKM120 or BYL719) has been investigated (NCT02088684) and those results are eagerly awaited to investigate first the MTD and then the efficacies of these combinations.

### 7.2. CDK4/6 and Immunotherapy Combinations

CDK4/6 inhibitors do not only induce cell cycle arrest in tumor cells but are also capable of eliciting an anti-tumor immune response. There are different mechanisms that have evinced such an immune response, including enhanced antigen presentation by tumor cells [82], stimulation of effector T lymphocyte activation [83], and reduction of proliferation of immunosuppressive T_reg_ cells [82]. The enhanced immune response upon CDK4/6 inhibition mainly involves the pathway of immune checkpoints [82,83]. Teo and colleagues showed that a combination of CDK4/6 and PD1 blockade resulted in a synergistic inhibition of tumor growth [74]. They showed that the combination of CDK4/6 inhibitors with PI3Kα inhibitors and immune checkpoint inhibitors (targeting PD-1 and CTLA-4) induced complete and durable regressions (>1 year) in established xenograft mouse models of human TNBC [74]. Immunotherapy is a frontline therapy that aims to improve outcomes of advanced cancer patients, and better biomarkers are needed in order to predict the outcome of therapy. Actually, various clinical studies aimed at identifying suitable biomarkers that could predict the effect of immunotherapy before giving it to patients in order to be able to better choose patients who would best respond to immune checkpoint inhibitors. Of note, data on the efficacy of combining CDK4/6 inhibitors and immune therapies come from various mouse models of BC. Preliminary positive results from a phase Ib clinical trial investigating abemaciclib with pembrolizumab, an anti-PD1 antibody, in ER-positive HER2-negative MBC demonstrated that this combination is safe and showed an ORR of 14.3% at a 16-week interim analysis [84]. Randomized placebo-controlled clinical trials investigating the effect of the inhibition of CDK4/6 and immune checkpoint inhibitors are now needed to shed more light on the efficacy of this combinatorial strategy in advanced BC patients.

## 8. Biomarkers

### 8.1. Clinical Parameters

The impact of clinical factors as predictors of responsiveness to abemaciclib was investigated by a combined analysis of MONARCH-2 and MONARCH-3 studies including over 1000 patients [85]. This analysis confirmed that the following factors have prognostic value: liver metastases, bone-only disease, tumor grade, progesterone receptor (PR) expression, and ECOG performance status. A poor prognosis was observed in patients who had liver metastases, a PR-negative status, and high-grade tumors. The analysis showed that the subgroups of patients who had a poor prognosis benefited the most from abemaciclib, and such an improvement was characterized by an improvement of mPFS (HR, 0.4 to 0.5) and ORR (more than 30%). Therefore, such clinical factors are an indicator that may help to guide clinicians to add abemaciclib to endocrine therapy [85].

### 8.2. Molecular Biomarkers

#### 8.2.1. Retinoblastoma Protein Expression

Rb has an important role in the CDK4/6 molecular pathway and consequently also in mediating anti-tumor responses to CDK4/6 inhibitors in breast cancer [41,46,86]. For this reason, it was initially postulated that tumors with alterations on the Rb molecule would be less likely to respond to CDK4/6 inhibition. Additionally, emerging clinical reports elicited that acquired resistance to CDK4/6 inhibitors correlates with novel polyclonal mutations in RB1 (encoding for the Rb protein) conferring loss of functions [87]. However, despite these evidence, several analyses of biomarkers from randomized PALOMA-2 and PALOMA-3 clinical trials failed to show a statistically significant association between the RB1 gene or Rb protein and a benefit deriving from CDK4/6 inhibitors [88,89,90]. A possible explanation is that acquired resistance to CDK4/6 occurs later on as a consequence of acquired RB1 mutations. Moreover, a phase I clinical trial including 27 patients expressing the Rb protein with advanced breast cancer has recently shown that the combination of palbociclib and paclitaxel is relatively safe for this subgroup. The clinical benefit rate at RP2D was 55% and it was observed across all receptor subtypes [91]. More investigations are warranted to evince if there is actually a role for the Rb gene and/or its acquired mutations in conferring a benefit to the use of CDK 4/6 inhibition.

#### 8.2.2. Cyclin D1 or p16^Ink4a^ Alterations

Initially, it was hypothesized that the amplification of *CCDN1* could make cancers more dependent on the CDK4/6 pathway and therefore more vulnerable to strategies of CDK4/6 inhibition. However, data from two randomized open-label and multi-centered clinical trials (PALOMA-1 and PALOMA-2) did not show any correlation between the levels of *CCND1* and responsiveness to palbociclib. In fact, alterations in the *CCND1* gene or the p16^Ink4a^ protein were neither in the phase II PALOMA-1 study [42], nor in the phase III PALOMA-2 trial [90] predictors of better survival for breast cancer patients. Therefore, there is a discrepancy between in vitro data, showing that low levels of p16^Ink4a^ are predictors for benefit from CDK4/6 inhibition, and data from clinical trials showing that such a correlation does not exist in patients [41,42,92,93].

#### 8.2.3. Cyclin E1 Alterations

Genetic analysis of tumor samples from the randomized clinical trial PALOMA-3 (fulvestrant treatment with palbociclib or placebo) showed that higher levels of the *CCNE1* gene (encoding for cyclin E1) correlated with relative resistance to palbociclib [88]. In fact, mPFS in patients with low levels of *CCNE1* (i.e., below median) was 14.1 months in the palbociclib group compared to 4.8 months in the placebo group (HR, 0.32). On the contrary, the mPFS of patients with high *CCNE1* levels (i.e., above the median) was 7.6 months in the palbociclib group compared to 4.0 months in the placebo group (HR, 0.85). Hence, palbociclib addition showed more benefit to patients with low *CCNE1* levels (*p* = 0.0024) [88]. Therefore, targeting CDK2 could be an alternative to overcome the resistance to palbociclib. However, since there are no other clinical trials reporting the same intriguing results, there is clearly a need for validation of the value of CDK2 in larger randomized clinical trials.

## 9. Conclusions

According to the clinical data from the use of CDK4/6 inhibitors in breast cancer, there have been significant improvements in the median PFS in breast cancer patients. These results led to FDA approval for the use of CDK4/6 inhibitors in combination with endocrine therapy for the treatment of ER-positive HER2-negative advanced breast cancer patients. Encouraging results also came from the combination of these drugs with other compounds in other BC subtypes, such as with anti-HER2 drugs in HER2-positive BC. The emerging data also suggest a potential use of CDK4/6-targeted approaches in neoadjuvant settings. Comparing palbociclib, ribociclib, and abemaciclib, there is no clear evidence demonstrating the superiority of one of these drugs. According to a recent meta-analysis including 3743 patients, the grade 3–4 toxicity profiles were similar between the three drugs. However, there was a lower risk of diarrhea for palbociclib compared to abemaciclib (relative risk, 0.13; 95% CI, 0.02 to 0.92; *p* = 0.04) as well as a reduced risk of prolonged QTc for palbociclib versus ribociclib (relative risk, 0.02; 95% CI, 0 to 0.83; *p* = 0.03) [94]. In conclusion, the three drugs are equally effective for the treatment of ER-positive HER2-negative breast cancer. Regarding biomarkers of responsiveness to CDK4/6 inhibitors, there is a need for a better understanding, which could come from ongoing clinical trials. To date, promising biomarkers for the prediction of treatment efficiency seem to be those subgroups of patients who had a poor prognosis as they were shown to benefit the most from abemaciclib. On the other hand, high levels of the *CCNE1* gene could be a potential biomarker for resistance to palbociclib. Moreover, an intriguing strategy could be that of inhibiting CDK2 concomitantly to the inhibition of CDK4/6 to improve the efficacy of the latter therapy.

## Figures and Tables

**Figure 1 cells-08-00321-f001:**
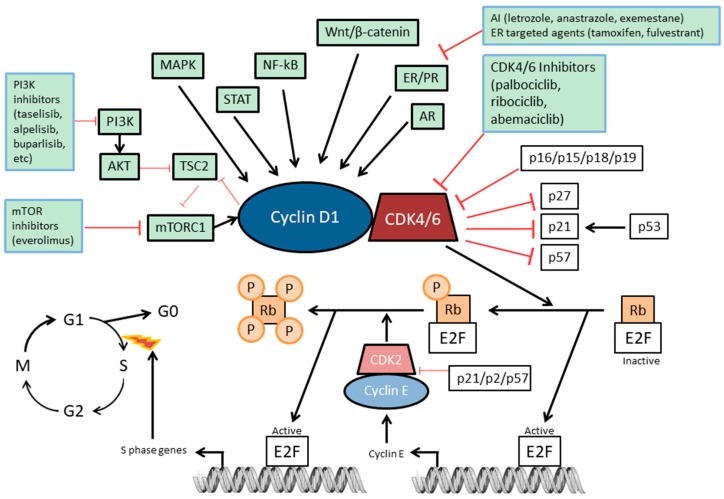
Cyclin D1–CDK4/6–Rb pathway in breast cancer. The molecular pathway and the major interconnected molecules are indicated. The inhibitory strategies that are most feasible in combination with CDK4/6 inhibition are indicated.

**Table 1 cells-08-00321-t001:** Cyclin-Dependent Kinases (CDK)4/6 inhibitors’ toxicity profiles.

	Palbociclib			Ribociclib			Abemaciclib		
	Monotherapy [46]	+letrozole PALOMA-2 [47]	+fulvestrant PALOMA-3 [48]	Monotherapy [49]	+letrozole MONALEESA-2 [43]	+fulvestrant MONALEESA-3 [50]	Monotherapy MONARCH-1 [44]	+fulvestrant MONARCH-2 [45]	+letrozole/anastrozole MONARCH-3 [51]
**Adverse Event (any grade)**	Neutropenia (92%)	Neutropenia (80%)	Neutropenia (80%)	Neutropenia (46%)	Neutropenia (75%)	Neutropenia (70%)	Neutropenia (85%)	Neutropenia (46%)	Neutropenia (41%)
	Leukopenia (100%)	Leukopenia (40%)	Leukopenia (50%)	Leukopenia (43%)	Leukopenia (33%)	Nausea (45%)	Leukopenia (90%)	Diarrhea (86%)	Diarrhea (81%)
	Thrombocytopenia (76%)	Nausea (35%)	Infections (42%)	Thrombocytopenia (30%)	Nausea (50%)	Fatigue (32%)	Thrombocytopenia (76%)	Nausea (45%)	Nausea (39%)
	Anemia (70%)	Fatigue (35%)	Fatigue (40%)	Fatigue (45%)	Infections (50%)	Diarrhea (29%)	Anemia (69%)	Fatigue (40%)	Fatigue (40%)
	Lymphopenia (65%)	Arthralgia (33%)	Nausea (32%)	Nausea (42%)	Fatigue (37%)	Leukopenia (28%)	Diarrhea (90%)	Abdominal pain (35%)	Infections (39%)
		Alopecia (33%)			Diarrhea (35%)	Vomiting (27%)	Fatigue (65%)		
					Alopecia (33%)	Constipation (25%)	Nausea (65%)		
						Arthralgia (24%)	Decreased appetite (45%)		
						Cough (22%)	Abdominal pain (40%)		
						Headache (22%)	Vomiting (35%)		
						Alopecia (19%)			
						Rash (18%)			
						Anemia (17%)			
**Adverse Event (grade 3–4)**	Neutropenia (54%)	Neutropenia (66%)	Neutropenia (65%)	Neutropenia (27%)	Neutropenia (59%)	Neutropenia (53%)	Neutropenia (27%)	Neutropenia (24%)	Neutropenia (21%)
	Leukopenia (51%)	Leukopenia (25%)	Leukopenia (28%)	Leukopenia (17%)	Leukopenia (21%)	Leukopenia (14%)	Leukopenia (28%)		
	Lymphopenia (30%)			Fatigue (2%)		Anemia (3%)	Diarrhea (20%)		
				Nausea (2%)		Fatigue (2%)			
						Back Pain (2%)			
						Nausea (1%)			
						Constipation (1%)			
						Headache (1%)			

**Table 2 cells-08-00321-t002:** Ongoing randomized phase II/III clinical trials investigating CDK4/6 inhibitors in breast cancer.

Study	Phase	Arms	Study Population	Primary Endpoint	Available Results
PENELOPE-B (NCT01864746) is an open-label, randomized clinical investigation, with 1250 participants	3	Adjuvant ET ± palbociclib in a 28-day cycle for 13 cycles	ER+/HER2− patients with residual disease and high risk of relapse after more than 16 weeks of neoadjuvant CT	iDFS	No results posted
PASTOR (NCT02599714) is an international, multicenter, randomized clinical trial with 54 participants	1b/2	Vistusertib + Palbociclib + Fulvestrant Placebo + Palbociclib + Fulvestrant	ER+ locally advanced or MBC postmenopausal patients pretreated with hormonal therapy	PFS	No results posted
PALOMA-2 (NCT01740427) is a randomized, multicenter clinical trial with 666 participants	3	Palbociclib + Letrozole Placebo + Letrozole	Postmenopausal women with ER+/HER2− advanced BC not previously treated	PFS	24.8 months (22.1 to NA*) in Palbociclib + Letrozole arm; 14.5 months (12.9 to 17.1) in Placebo + Letrozole arm. * The value was not available because there were not enough disease progression events in the treatment group at the time of analysis, due to drug benefit.
NEOPAL (NCT02400567) is an open-label, multicenter, international, randomized clinical trial with 125 participants	2	3_FEC (Fluorouracil-Epirubicin-Cyclophosphamide) Letrozole + Palbociclib	Luminal BC in postmenopausal women	RCB	No results posted
PALLET (NCT02296801) is an open-label, randomized clinical trial with 306 participants	2	Letrozole Palbociclib + Letrozole Letrozole then Letrozole + Palbociclib Palbociclib then Letrozole + Palbociclib	Early invasive BC with ER+/HER2− not previously treated	Ki-67 proliferation cCR	No results posted
PALOMA-3 (NCT01942135) is a double-blind, randomized, placebo-controlled clinical trial with 521 participants	3	Palbociclib + Fulvestrant Placebo + Fulvestrant	ER+/HER2− MBC with progression after prior endocrine therapy	PFS	9.2 months (7.5 to NA*) in Palbociclib + Fulvestrant arm; 3.8 months (3.5 to 5.5) in Placebo + Fulvestrant arm. * This parameter is not estimable when the Kaplan Meier-based curve representing the upper confidence limits for survival function lies above 50%.
PALOMA-4 (NCT02297438) is a multicenter, randomized, double-blind clinical trial with 339 participants	3	Palbociclib + Letrozole Placebo + Letrozole	Untreated Asian postmenopausal women with ER+/HER2− advanced BC	PFS	No results posted
PEARL (NCT02028507) is an international, open-label, randomized clinical trial with 596 participants	3	Palbociclib + Exemestane or Fulvestrant Capecitabine	ER+/HER2− MBC with resistance to aromatase inhibitors	PFS	No results posted
NCT02630693 is an open-label, randomized clinical trial with 180 participants	2	Palbociclib 100 mg daily + Fulvestrant or Tamoxifen or another aromatase inhibitor Palbociclib 125 mg daily + Fulvestrant or Tamoxifen or another aromatase inhibitor	ER+/HER2− advanced or MBC	PFS	Results submitted but not publicly available
PARSIFAL (NCT02491983) is an open-label, multicenter and randomized clinical trial with 486 participants	2	Palbociclib + Fulvestrant Palbociclib + Letrozole	ER+/HER2− MBC	PFS	No results posted
MONALEESA-2 (NCT01958021) is randomized, double-blind and placebo-controlled clinical trial with 670 participants	3	Ribociclib + Letrozole Placebo + Letrozole	Postmenopausal women with ER+/HER2− advanced BC with no prior therapy	PFS	NA* (19.3 months to NA) in LEE011 + Letrozole arm; 14.7 months (13.0 to 16.5) in Placebo + Letrozole arm. * N/A = not estimable as median PFS was not reached in the ribociclib arm.
MONALEESA-3 (NCT02422615) is a double-blind, placebo-controlled and randomized clinical trial with 780 participants	3	Ribociclib + Fulvestrant Placebo + Fulvestrant	Postmenopausal women and men with ER+/HER2− advanced BC with only 1 line or no prior endocrine therapy	PFS	20.5 months (18.5 to 23.5) in Ribociclib + Fulvestrant arm. 12.8 months (10.9 to 16.3) in Placebo + Fulvestrant arm.
MONALEESA-7 (NCT02278120) is a multicenter, double-blind, placebo-controlled and randomized clinical trial with 672 participants	3	Ribociclib + NSAI or Tamoxifen + Goserelin Placebo + NSAI or Tamoxifen + Goserelin	Premenopausal women with ER+/HER2− advanced BC	PFS	23.8 * months (19.2 to NA) in LEE011 + NSAI/Tamoxifen + Goserelin arm; 13.0 months (11.0 to 16.4) in Placebo + NSAI/Tamoxifen + Goserelin arm. * N/A = The value was not available due to insufficient number of participants with events in the treatment arm to enable estimation of the upper limit of the confidence interval.
MONARCH2 (NCT02107703) is a randomized, double-blind, placebo-controlled clinical trial with 669 participants	3	Abemaciclib + Fulvestrant Placebo + Fulvestrant	ER+/HER2− locally advanced or MBC	PFS	16.4 months (14.4 to 19.3) in Abemaciclib + Fulvestrant arm; 9.3 months (7.4 to 11.4) in Placebo + Fulvestrant arm
MONARCH3 (NCT02246621) is a randomized, double-blind, placebo-controlled clinical trial with 493 participants	3	Abemaciclib + NSAI Placebo + NSAI	ER+/HER2− locoregionally recurrent or MBC not previously treated in postmenopausal women	PFS	NA * (NA to NA) in Abemaciclib + NSAI arm. * (Median of PFS not reached due to a number of events not yet attained); 14.73 months (11.11 to 17.46) in Placebo + NSAI arm.
MONARCHplus (NCT02763566) is a randomized, double-blind, placebo-controlled clinical trial with 450 participants	3	Abemaciclib + NSAI Placebo + NSAI Abemaciclib + Fulvestrant Placebo + Fulvestrant	ER+/HER2− locoregionally recurrent or MBC postmenopausal women	PFS	No results posted
nextMONARCH 1 (NCT02747004) is a randomized, open-label clinical trial with 225 participants	2	Abemaciclib + Tamoxifen Abemaciclib Abemaciclib + Prophylactic Loperamide	ER+/HER2− MBC previously treated	PFS	No results posted
monarcHER (NCT02675231) is an open-label, randomized clinical trial, with 225 participants	2	Abemaciclib + trastuzumab + fulvestrant Abemaciclib + trastuzumab Trastuzumab + CT	ER+/HER2− locally advanced or metastatic breast cancer patients	PFS	No results posted

Abbreviations: BC, breast cancer; MBC, metastatic breast cancer; HR+, hormone receptor positive; HER2-, human epidermal receptor 2 negative; CT, chemotherapy; ET, endocrine therapy; iDFS, invasive disease-free survival; PFS, progression-free survival; RCB, residual cancer burden; cCR, clinical complete response; NSAI, non-steroidal aromatase inhibitor.

**Table 3 cells-08-00321-t003:** Recruiting randomized phase II/III clinical trials investigating CDK4/6 inhibitors in breast cancer.

Study	Phase	Arms	Study Population	Primary Endpoint
PALLAS (NCT02513394) is an open-label, randomized clinical trial, with an estimated number of 4600 patients.	3	Adjuvant ET (≥5 years) ± palbociclib (for 2 years)	ER+/HER2− Patients at stage II or III	iDFS
PATRICIA (NCT02448420) is an open-label, randomized clinical trial, with an estimated number of 138 patients	2	Palbociclib + trastuzumab ± letrozole Palbociclib + trastuzumab	HER2+/ER± locally advanced or metastatic BC post-menopausal women treated previously with CT and trastuzumab	PFS
PATINA (NCT02947685) is an open-label, randomized clinical trial, with an estimated number of 496 patients	3	Palbociclib + anti-HER2 + endocrine therapy	HER2+/ER+ MBC patients who received anti-HER2-based induction of CT before enrollment	PFS
NCT02592746 is an open-label, randomized clinical trial, with an estimated number of 182 patients	2	Palbociclib + Exemestane + GnRH agonist Capecitabine	ER+ Premenopausal Women with MBC	PFS
PADA-1 (NCT03079011), is an open-label, randomized clinical trial, with an estimated number of 800 patients	3	A: Palbociclib + AI (Letrozole, Anastrozole, Exemestane) B: Palbociclib + Fulvestrant Selection: Palbociclib + AI (Letrozole, Anastrozole, Exemestane)	ER+/HER2− MBC	Safety until randomization Efficacy from randomization
T-DM1 (NCT03530696) is an open-label, randomized clinical trial, with an estimated number of 132 patients	2	T-DM1 + Palbociclib T-DM1	HER2+ MBC	PFS
SAFIA (NCT03447132) is a multicenter, international, randomized clinical trial, with an estimated number of 400 patients	3	Fulvestrant + Palbociclib Fulvestrant + Placebo	Operable Luminal BC responding to Fulvestrant	pCR
PACE (NCT03147287) is an open-label, randomized clinical trial, with an estimated number of 220 patients	2	Fulvestrant Fulvestrant + Palbociclib Palbociclib + Fulvestrant + Avelumab	Metastatic ER+/HER2− BC previously stopped responding to palbociclib and endocrine therapy	PFS
NCT02384239 is an open-label, randomized clinical trial, with an estimated number of 70 patients	2	Palbociclib 100 mg plus fulvestrant or tamoxifen Palbociclib 125 mg + fulvestrant or tamoxifen	ER+ in MBC patients previously exposed to inhibitors of the PI3K Pathway	Tumor progression
PASIPHAE (NCT03322215) is an open-label, multicenter international and randomized clinical trial, with an estimated number of 196 patients	2	Palbociclib + Fulvestrant Capecitabine	Metastatic ER+/HER2− BC with progressive disease after endocrine treatment	PFS
PATHWAY (NCT03423199) is a multicenter, international, randomized clinical trial, with an estimated number of 180 patients	3	Palbociclib + Tamoxifen ± Goserelin Placebo + Tamoxifen ± Goserelin	ER+/HER2− Advanced or metastatic breast cancer patients, regardless of menopausal status	PFS
NCT02913430 is an open-label, randomized clinical trial, with an estimated number of 150 patients	2	Fulvestrant + Palbociclib Tamoxifen + Palbociclib	ER+ with MBC with 2 or 3 prior lines of endocrine therapy and up to one line of CT for MBC	PFS
ImmunoADAPT (NCT03573648) is an open-label, randomized clinical trial, with an estimated number of 40 patients	2	Tamoxifen + Avelumab Tamoxifen + Palbociclib + Avelumab	Early Stage ER+ BC	cCR
PELOPS (NCT02764541) is an open-label, randomized neoadjuvant clinical trial, with an estimated number of 180 patients	2	Tamoxifen + Endocrine Therapy Letrozole + Endocrine Therapy Tamoxifen + Endocrine Therapy + Palbociclib Letrozole + Endocrine therapy + Palbociclib	Invasive Lobular Carcinoma and Invasive Ductal Carcinoma	Anti-proliferative activity (Ki67) pCR
MORPHEUS (NCT03280563) is an open-label, multicenter, randomized clinical trial, with an estimated number of 111 patients	1b/2	Fulvestrant Atezolizumab + Entinostat Atezolizumab + Fulvestrant Atezolizumab + Ipatasertib Atezolizumab + Ipatasertib+ Fulvestrant Atezolizumab + Bevacizumab + Endocrine therapy Mandatory on-treatment biopsy	ER+/HER2− BC with progression during or following treatment with CDK 4/6 inhibitor in the first or second line setting	% of patients with objective response
MAINTAIN (NCT02632045) is a randomized clinical trial, with an estimated number of 132 patients	2	Ribociclib + Fulvestrant Placebo + Fulvestrant	Metastatic ER+/HER2− BC with progression after Aromatase Inhibitor + CDK4/6 therapy	Percentage of progression-free at 24 weeks
KENDO (NCT03227328) is an open-label, multicenter, randomized clinical trial, with an estimated number of 150 patients	2	CDK4/6 Inhibitor + Endocrine therapy Standard CT + Endocrine therapy	Advanced ER+/HER2− BC	PFS
PREDIX LumB (NCT02603679) is an open-label, randomized clinical trial, with an estimated number of 200 patients	2	Paclitaxel Tamoxifen + Palbociclib Aromatase In + Palbociclib Goserelin + Aromatase In + Palbociclib	ER+ tumors with high proliferation or low proliferation with metastatic nodes	Radiological objective response rate
PREDIX LumA (NCT02592083) is an open-label, randomized clinical trial, with an estimated number of 200 patients	2	Endocrine treatment Endocrine treatment + Palbociclib	Early stage ER+ BC	Clinical and radiological response
SONIA (NCT03425838) is an open-label randomized clinical trial, with an estimated number of 1050 patients	3	Non-steroidal aromatase Inhibitor + CDK4/6 inhibitor in first line + Fulvestrant in the 2^nd^ line Non-steroidal aromatase Inhibitor + Fulvestrant + CDK4/6 inhibitor in the 2nd line	ER+/HER2− advanced BC	PFS (after 2 lines of treatment)
NCT03128619 is an open-label, randomized clinical trial, with an estimated number of 102 patients	1b/2	Copanlisib + Letrozole Copanlisib + Letrozole + Palbociclib	ER+/HER2− Postmenopausal Any stage BC	Change in Ki-67 expression The incidence of dose-limiting toxicities (DLT)
FLIPPER (NCT02690480)is an international, multicenter, randomized clinical trial, with an estimated number of 190 patients	2	Palbociclib + Fulvestrant Placebo + Fulvestrant	Postmenopausal women with ER+/HER2− MBC after 5 or more years of endocrine therapy in adjuvant setting	PFS
TOUCH (NCT03644186) is an open-label, international neoadjuvant clinical trial, with an estimated number of 144 patients	2	Paclitaxel + Trastuzumab and Pertuzumab Palbociclib + Letrozole + Trastuzumab and Pertuzumab	ER+/HER2+ early BC	pCR
NEOLBC (NCT03283384) is an open-label randomized clinical trial, with an estimated number of 100 patients	2	Letrozole Standard CT Ribociclib + Letrozole	Postmenopausal ER+/HER2− Stage II/III Luminal BC	Difference in complete cell cycle arrest (Ki-67 < 1%)
RIBBIT (NCT03462251) is a prospective, randomized, open-label multicenter clinical trial, with an estimated number of 160 patients	3	Ribociclib + Aromatase Inhibitor Paclitaxel ± Bevacizumab	Postmenopausal women ER+/HER2− Advanced BC with visceral metastases and no prior therapy for advanced disease	PFS
AMICA (NCT03555877) is a multicenter, prospective, randomized, open-label clinical trial, with an estimated number of 150 patients	2	Anti-hormonal treatment + ribociclib Anti-hormonal treatment	ER+/HER2− BC patients with disease control (at least stable disease) after 1^st^ line CT	PFS
NCT03671330 is a randomized, placebo-controlled clinical trial, with an estimated number of 315 patients	2	Ribociclib Ribociclib Placebo	Chinese postmenopausal women with ER+/HER2− Advanced BC	PFS
CORALLEEN (NCT03248427) is a two-arm, multicentric, randomized clinical trial, with an estimated number of 94 patients	2	Ribociclib + Letrozole CT (doxorubicin and cyclophosphamide) + paclitaxel	Postmenopausal women with primary operable Luminal B ER+/HER2− BC	Residual Cancer Burden (RCB)
FELINE (NCT02712723) is a randomized clinical trial, with an estimated number of 120 patients	2	Placebo + Letrozole Ribociclib + Letrozole	ER+/HER2− Early BC	Rate of Pre-operative Endocrine Prognostic Index (PEPI) score 0 at surgery
LEADER (NCT03285412) is an open-label randomized clinical trial, with an estimated number of 120 patients	2	Ribociclib Intermittent (21 days) + Endocrine Therapy daily Ribociclib (28 days) + Endocrine Therapy daily	ER+ Early stage BC	Number of patients who complete 12 months of treatment
NCT03703466 is an open-label, randomized clinical trial, with an estimated number of 60 patients	2	Abemaciclib + a meal Abemaciclib without a meal Abemaciclib without regard to food	ER+/HER2− MBC previously treated	% of participants with severe diarrhoea with grade 3, 2 % of participants with dose reduction, dose interruption and discontinue treatment due to diarrhoea % of participants utilizing antidiarrheal/s
MonarchE (NCT03155997) is an open-label, randomized clinical trial, with an estimated number of 3580 patients	3	Adjuvant ET ± abemaciclib	ER+/HER2− Node-positive high-risk breast tumors (≥ 4 lymph nodes, tumor > 5 cm, grade 3 or central Ki67 ≥ 20%)	iDFS

Abbreviations: BC, breast cancer; MBC, metastatic breast cancer; HR+, hormone receptor positive; HER2−, human epidermal receptor 2 negative; cCR, clinical complete response; CT, chemotherapy; ET, endocrine therapy; iDFS, invasive disease-free survival; pCR, pathological complete response; PFS, progression-free survival.

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
