# Peer review of "Updates on the CDK4/6 Inhibitory Strategy and Combinations in Breast Cancer"

_cells, 2019, doi:10.3390/cells8040321_

Round 1
Reviewer 1 Report
The authors review the current molecular background and clinical efficacy of CDK4/6 inhibitors for breast cancer treatment. Particularly, these inhibitors are currently used for the treatment of estrogen receptor (ER)-positive breast cancer. In this review, the authors also introduce to the future directions of ongoing clinical trials, which are also investigating the use of CDK4/6 inhibitors in HER-2-positive breast cancer, and also debate about predictive biomarkers of CDK4/6 inhibitors sensitivity. This review is of high interest for clinicians specialized in treatment of breast cancer worldwide. The article summarizes the clinical trials comparing efficacy of the currently FDA-approved CDK4/6 inhibitors in different breast cancer subtypes and also discusses their safety, tolerability and adverse effects. These family of inhibitors constitute the most up-to-date targeted therapy in breast cancer, which may offer a new therapeutic tool not only for ER-positive breast cancer but also for other breast cancer subtypes.
I only have some minor comments.
1- This referee believes that Figure 1 should be modified in order to include a more detailed model of the RB1 pathway (section #2, page 2, line 78) and of the mechanisms of action of CDK4/6 inhibitors (section #3, page 2, line 88). The RB1 pathway is described in the manuscript (Page 2, line 78). Hence, no citation of the figure 1 is mentioned in relation to that paragraph. On the other hand, some of the proteins described in the text along section #3 (such as p27 or p57) are not present in Figure 1 and some modulators indicated in Figure 1 (such as MAPK, Wnt/beta-catenin pathway, AR, STAT, NFkB) are not described in the manuscript. This referee believes that the quality of the manuscript could be improved by including this information.
2- Palbociclib, ribociclib, and abemaciclib are selective reversible inhibitors of CDK4/6. This reviewer believes that the word “reversible” should be included in the first description of these agents, since this is an important pharmacological datum.
3- In page #15, line 256, please include neratinib in the list of HER2 kinase inhibitors. Neratinib is the latest HER2 kinase inhibitor approved by FDA (https://www.fda.gov/drugs/informationondrugs/approveddrugs/ucm567259.htm)
4- In section #7.1: “ CDK4/6 and PI3K-mTOR inhibitions”, the authors indicate: “There is a crosstalk between the CDK4/6 and the PI3K-mTOR pathways, resulting in a good rationale for testing the combined inhibition of the two pathways in order to block tumor growth [71,82–84].” This referee believes that a brief description of how this crosstalk occurs should be included in the manuscript to help the reader understand the molecular rationale behind their combination.
Author Response
Reviewer#1
Comment#1:
We modified Figure 1 to include additional details regarding Rb pathway and CDK4/6 inhibitors that we mentioned in the text. We added references to Figure 1 at several positions (line 51, 80, 90). In addition, we added a sentence at the beginning of the section about the RB1 pathway (line 79) indicating that D-type cyclins are regulated by several signaling pathways "In response to various external stimuli, a number of signaling pathways (e.g., PI3K/AKT/mTOR, MAPK, STAT, NF-kB, Wnt/beta-Catenin, ER, PR, AR) affect the expression and stability of D-type cyclins."
Comment#2:
We added "reversible" in line 103 as suggested by the reviewer
Comment#3:
We added "neratinib" in line 256 as suggested by the reviewer
Comment#4:
We added 1-2 sentences regarding the crosstalk between CDK4/6 and PI3K-mTOR pathways: <<In fact, Goel et al. evinced that inhibition of CDK4/6 not only suppressed Rb phosphorylation, but also reduced the TSC2 phosphorylation thereby partially hindering the mTORC1 activity. In line with this, in vivo results proved that CDK4/6 sensitized PDX to HER2-targeted therapies [71]. Additionally, Vora et al. demonstrated through a screening of 42 CDK4/6 inhibitors that they were able to sensitize PI3KCA mutation-bearing breast cancer cell lines to PI3K inhibitors. The strongest sensitizing CDK4/6 inhibitor was ribociclib. Furthermore they confirmed this data using PIK3CA mutation-bearing mouse xenografts, in which co-treatment with CDK4/6 and PI3K inhibitors elicited tumor regression [83].>>
Reviewer 2 Report
This review article by Sobhani et al. aims to summarizes the molecular mechanisms of action and the clinical use of the three FDA-approved CDK4/6 inhibitors for the treatment of ER+/HER-breast cancers and their potential future use in other molecular subtypes of breast cancer. This well-written manuscript highlights the major clinical trials that have been conducted to test efficacies of the CDK4/6 inhibitor alone and in combination therapies. The primary critique of this manuscript is the way the data is visually presented in the three tables used to summarize data, which may be adjusted during publication formatting. In particular, Table 2 would benefit from having lines separating the columns that describe the three drugs and to “top align” the rows so that it is easier to read across and delineate the results of the study. In addition, adding another row to this table summarizing the clinical results (i.e.: mPFS or OS) from the presented studies would greatly help the reader compare the three drugs.
Author Response
Reviewer#2:
Comment#1:
The reviewer suggested that "Table 2 would benefit from having lines separating the columns ...". We think he meant Table 1. We prepared in the manuscript a version of Table 1 with the separated columns.
Comment#2:
The reviewer also suggested "adding another row to this table summarizing the clinical results ..." Since Table 2 summarized the ongoing clinical trials and not all the results of such investigations are available yet, we added the results of only those clinical trials whose results are available at the current time (8/17). As a consequence we added the preliminary results from the ongoing clinical trials also in the text of the manuscript; we added the data from PALOMA 3 in the Palbociclib chapter and the data from Monaleesa-7 in the Ribociclib chapter. Additionally, we consequently saw that it needs to be added two short paragraphs on the combination of palbociclib with enzalutamide in Rb proficient and AR positive TNBC cell lines and the recently published data of a phase I clinical trial testing palbociclib and paclitaxel in patients expressing Rb protein in chapter 8.2.1. We made this minor correction too.